# Pyrolysis of High-Ash Natural Microalgae from Water Blooms: Effects of Acid Pretreatment

**DOI:** 10.3390/toxins13080542

**Published:** 2021-08-03

**Authors:** Longfei Liu, Yichen Liu, Wenli Wang, Yue Wang, Guiying Li, Changwei Hu

**Affiliations:** Key Laboratory of Green Chemistry and Technology, Ministry of Education, College of Chemistry, Sichuan University, 29 Wangjiang Road, Chengdu 610064, China; liulongfei@stu.scu.edu.cn (L.L.); liuyc@stu.scu.edu.cn (Y.L.); wangwenli30@163.com (W.W.); Yue_Wang@stu.scu.edu.cn (Y.W.)

**Keywords:** microalgae, pretreatment, pyrolysis, de-ashing, bio-oil

## Abstract

Natural microalgae (NA, cyanobacteria) collected from Taihu Lake (Jiangsu, China) were used for biofuel production through pyrolysis. The microalgae were de-ashed via pretreatment with deionized water and hydrochloric acid, and the samples obtained were noted as 0 M, 0.1 M, 1 M, 2 M, 4 M, 6 M, 8 M, respectively, according to the concentration of hydrochloric acid used in the pretreatment. Pyrolysis experiments were carried out at 500 °C for 2 h. The products were examined by various techniques to identify the influence of the ash on the pyrolysis behavior. The results showed that the ash inhibited the thermal transformation of microalgae. The 2 mol/L hydrochloric acid performed the best in removing ash and the liquid yield increased from 34.4% (NA) to 40.5% (2 M). Metal-oxides (mainly CaO, MgO, Al_2_O_3_) in ash promoted the reaction of hexadecanoic acid and NH_3_ to produce more hexadecanamide, which was further dehydrated to hexadecanenitrile. After acid pretreatment, significant improvement in the selectivity of hexadecanoic acid was observed, ranging from 22.4% (NA) to 58.8% (4 M). The hydrocarbon compounds in the liquid product increased from 12.90% (NA) to 26.67% (2 M). Furthermore, the acid pretreatment enhanced the content of C_9_–C_16_ compounds and the HHV values of bio-oil. For natural microalgae, the de-ashing pretreatment before pyrolysis was essential for improving the biocrude yield and quality, as well as the biomass conversion efficiency.

## 1. Introduction

A lot of scientists are emphasizing the tremendous challenges that will soon be faced when the depletion of fossil fuel supplies coincides with an alarming increase in the global population. Growing concerns about the lack of fossil-based petroleum sources have promoted the utilization of renewable resources for fuels and value-added chemicals. Among various species of biomass, microalgae is considered as one kind of green, renewable and potential resource [1,2,3], owing to its fast growth rate, no occupation of arable land, high lipid content and efficient carbon dioxide fixation [4].

Taihu Lake is one of the five major freshwater lakes in China. Every year, cyanobacterial blooms and microcystins are produced in Taihu Lake, which seriously disrupts the ecological balance of the lake and severely affects the health of surrounding residents and livestock [5,6]. However, there is still no effective means to deal with large amounts of algae. If the abundant algae could be exploited to produce fuel precursors, it would provide not only a new promising method for biofuel production, but also a new way to reduce the damage to the environment.

The main microalgae conversion technologies used to obtain biofuels are transesterification and thermochemical routes. Transesterification is a method for reducing the molecular weight and the viscosity of the bio-oil. In the transesterification reaction, triglycerides and lower alcohols (usually methanol) are transesterified under the action of strong acids or strong bases to obtain fatty acid alkyl esters (i.e., biodiesel) and glycerol [7]. Thermochemical routes include pyrolysis and liquefaction [8,9]. Pyrolysis is the thermal decomposition of biomass in an atmospheric-pressure inert environment, usually at high temperatures (400–600 °C) [10]. Liquefaction can convert high-moisture biomass to biocrude in a water medium and thus does not require preliminary drying processes. It usually occurs with water under high pressure (5–25 MPa) and subcritical water temperature (280–370 °C) [11]. Among these conversion methods, pyrolysis is an effective technology for microalgae conversion to bio-oil [12,13] due to its low costs of installation and operation. After pyrolysis, solid, liquid, and gaseous products, namely bio-char, bio-oil, and bio-gas, respectively, can all be obtained. Among the three types of products, bio-oil is considered to be the most important product since it has the potential to replace fossil fuel. The bio-oil obtained by pyrolysis usually consists of O-compounds and N-compounds, thus causing its high acidity and corrosiveness [12]. Therefore, the direct utilization of bio-oil may be harmful to equipment and even cause secondary pollution to the environment [14,15]. There have been many studies [16,17,18,19] focused on acquiring high-quality bio-oil. Among these studies, pretreatment is one of the most effective methods for achieve this goal. A suitable pretreatment process of microalgae can increase the ease of use of the components for a wide range of applications, especially in the valorization of microalgae.

The pretreatment methods applied to microalgal biomass can be classified as follows: mechanical methods, thermal methods, chemical methods, and biological methods [16]. Among the four types of methods, chemical pretreatments are preferred because of their low energy requirements and easy scalability [20]. Meanwhile, chemical pretreatments can change the chemical structure of biomass and promote the pyrolysis process. Joonhyuk Choi et al. [21] reported that the pretreatment of brown algae with CaCl_2_ and sulfuric acid could remove the catalytically active inorganic minerals, soften the structure of brown algae, improve the performance of pyrolysis oil, and avoid the production of aggregated large particles of biochar. Zhou et al. [3] pretreated the natural microalgae by performing a hydrothermal treatment at 200 °C to extract sugar and part of the protein, simplifying the composition (mainly fatty acid ethyl ester), reducing the nitrogen content (<4%) and increasing the higher heating value (~41 MJ·kg^−1^) of the biocrude. Yu et al. [22] performed dilute sulfuric acid pretreatment on *Chlorella* at 160–170 °C to produce microalgae hydrolysate and biochar, and the microalgae hydrolysate obtained after pretreatment was totally composed of reducing sugars, with the highest concentration of 98.11 g/L. The maximum conversion rate was 95.22%.

Natural microalgae grow in different water environments (such as wastewater and eutrophication) and contains lower lipids and higher ash content. As reported in the literature [23,24], inorganic minerals in ash were able to increase the yield of carbon and oxygen-containing compounds from pyrolysis, which could cause higher acidity and lower stability of bio-oil. To overcome the side effects caused by high ash content, acid washing pretreatment was considered to be effective. Recently, Liu et al. [25] used 4 mol/L hydrochloric acid to pretreat natural microalgae with high ash content to obtain de-ashed microalgae at 25 °C, improving its behavior in hydrothermal conversion, where it was found that the hydrochloric acid pretreatment could increase the yield of bio-oil, while the calcium carbonate in the ash promoted deamination, resulting in an increase in the relative content of ketones in the light oil. Another study carried out by Liu et al. [26] where 0.1 mol/L hydrochloric acid was used to pretreat sewage sludge at ambient temperature. The results of catalytic pyrolysis showed that acid pretreatment reduced the catalytic effect of ash in sewage sludge and increased the production of hydrocarbons, where the highest yield of hydrocarbons achieved was 51.4% at 650 °C. There were few studies that focused on the effect of hydrochloric acid washing pretreatment on the pyrolysis of the high-ash natural microalgae, and the effect of different metal-oxides in the ash on the pyrolysis were also rarely reported.

In the present work, we studied the pretreatment of natural microalgae from Taihu Lake with different concentrations of hydrochloric acid to find the best concentration to remove ash and thus augment the yield and quality of bio-oil. The liquid products were detected and analyzed by GC-MS, GPC, and elemental analysis. The effects of calcium oxide, magnesium oxide, and aluminum oxide on the transformation of hexadecanoic acid (C_15_H_31_COOH) and hexadecanamide (C_15_H_31_CONH_2_) were investigated as well.

## 2. Results

### 2.1. The Ash Content and Metal Content of Microalgae Samples

In this work, the natural microalgae (NA) from Taihu Lake, mainly consisting of cyanobacteria *Microcystis*, were used as feedstock. The characteristics of the feedstock are shown in Table 1. The feedstock contains a lot of proteins (41.2 wt%), resulting in the high content of nitrogen. In addition, it also contains a certain amount of carbohydrates (10.4 wt%) and lipids (6.6 wt%). Compared to the lignocellulosic biomass, the ultimate analysis shows that the content of carbon and hydrogen in the feedstock is low, and the oxygen content is high, which makes the HHV of the microalgae to be only 14.9 MJ/kg. At the same time, because it grows in the natural environment, the ash content reaches up to 33.2 wt%. The fixed carbon content is 15.4 wt%. Therefore, the microalgae from Taihu Lake are not suitable for direct use.

As shown in Table 2, for the NA (natural microalgae) sample, the ash content was 33.2 wt%. With the concentration of hydrochloric acid rising, the ash content presented a trend of first declining and then ascending. Specifically, for the 0 M and 0.1 M samples, the ash content hardly changed. For the 2 M sample, the ash content decreased to 21.1 wt%. However, for the 4 M, 6 M, 8 M samples, the ash content increased to 30.9 wt%, 32.9 wt%, 43.3 wt%, respectively.

The metal content changes before and after the pretreatment with different concentrations of acid were shown in Figure 1. For the NA sample, the contents of K, Ca, Na, Mg, Al, and Fe were 3.22, 8.08, 1.77, 0.97, 36.82, 10.16 mg/g, respectively. With the concentration of hydrochloric acid increased from 0 mol/L to 8 mol/L, their content presented a trend of first decreasing and then increasing. Specifically, for the 2M sample, the contents of K, Ca, Na, Mg reached the lowest point, which were 1.18, 0.32, 0.38, 0.50 mg/g, respectively. However, further increasing the concentration of hydrochloric acid would cause their content to increase. As for the contents of Al and Fe, they reached their lowest points at 5.78 and 4.78 mg/g for the 4M sample. The content of Al was also augmented with further increase in the concentration of hydrochloric acid. These results indicated that too high a concentration of hydrochloric acid was not conducive to the removal of metals in ash. As for the content of Si, there was hardly any change in Si content. This indicated that the acid pretreatment removed most of the metals in microalgae, leaving only Si that would not participate in the reaction [25].

### 2.2. Dissolution Conversion of Carbohydrates

The contents of carbohydrates remaining in solid phase (the feedstocks) and dissolved in liquid phase (the pretreatment solutions) by the acid pretreatment, both based on the weight of raw microalgae or corresponding pretreated microalgae, were presented in Figure 2. From the content of carbohydrates in solid phase, it was found that minimal amounts of carbohydrates were dissolved for the 0 M, 0.1 M, 1 M and 2 M samples. However, the content of carbohydrates showed a sharp decrease for the 4 M, 6 M, and 8 M samples. This illustrated that when the concentration of hydrochloric acid was too high, it would dissolve more carbohydrates rather than removing metals in ash.

Similar to the solid phase, for the 0 M, 0.1 M, 1 M and 2 M samples, the content of carbohydrates in liquid phase was small and changed only a little. The content of carbohydrates in liquid phase increased remarkably for the 4 M, 6 M and 8 M samples. The results made sense for the ash content and metal content discussed in Section 2.1. When the concentration of hydrochloric acid was more than 2 mol/L, the dissolution of carbohydrates dominated in the pretreatment process, resulting in the increase in the ash content and metal content for the 4 M, 6 M, 8 M samples.

### 2.3. TG Analysis

The TG (thermogravimetric) and DTG (differential thermogravimetric) curves of solid samples are presented in Figure 3. As seen from the TG curves (Figure 3a), there were three thermal degradation stages in the pyrolysis, which was consistent with the findings of Hu et al. [27]. The first stage (40–200 °C) was mainly caused by the loss of water and light volatile compounds in microalgae. The weight loss of the second stage was about 50 wt% from 200 to 500 °C, and the third stage occurred from 500 to 800 °C.

As shown in Figure 3b, two major peaks were found in the second stage. The two peaks in this stage could be attributed to the difference between the thermal stable components of the feedstock. According to the literature [28,29], the first peak, between 250 and 290 °C, denoted the decomposition of carbohydrates and lipids. The second peak, between 320 and 380 °C, corresponded to the decomposition of proteins. For the 0 M sample, the first peak hardly changed. While for the 0.1 M, 1 M, 2 M, 4 M, and 6 M samples, the first peak all shifted to the left, indicating that by the acid pretreatment, the initial decomposition temperature would be decreased. The result indicated that the water pretreatment had little effect on the variation of pyrolysis temperature. However, for the 8 M sample, the first peak disappeared, owing to the removal of most carbohydrates in the microalgae.

As for the third stage (500–800 °C), it was mainly led by the slow decomposition of residual carbon and minerals [30]. For the production of residual carbon, NA, 0 M and 0.1 M samples showed nearly no difference, indicating that water or very dilute acid pretreatment had little effect on the yield of residual carbon (about 48%). With the increase of the HCl concentration in pretreatment, the residual carbon started to decrease and reached the least yield for 2 M sample (43.9%). However, as the HCl concentration further increased, the residual carbon started to rise and reached the highest yield for 8 M samples (54.4%). These results showed the same trend as the variation of ash content in Table 2, indicating the positive correlation between the yield of residual carbon and ash content, while the decrease of the peak temperature at 250–290 °C by dilute acid pre-treatment could be one of the reasons for the improved pyrolysis efficiency for 2 M sample. These results also indicated that pretreatment by 2 mol/L hydrochloric acid was the best for utilizing the pyrolysis process to obtain bio-oil with higher yield.

### 2.4. Yields of Pyrolysis Products

The liquid yield in NA samples was on average 34.2%, and there was almost no change when pretreated with water (0 M) or dilute acid (0.1 M) (Figure 4). When the concentration of hydrochloric acid was moderate (1 M, 2 M), the liquid yield was improved to some extent, and the highest liquid yield of 40.5% was achieved for the 2 M sample. However, for the 4 M, 6 M and 8 M samples, the liquid yield decreased to 38.4%, 34.3%, 28.5%, respectively. Therefore, in terms of achieving higher liquid yield, the 2 M sample performed the best.

The HHV of microalgae samples and bio-oil were analyzed, the results of which are listed in Appendix A. It could be seen that the HHV of microalgae samples and bio-oil were both improved to a certain degree. The highest HHV of bio-oil achieved was 30.4 MJ/kg for the 2 M sample.

### 2.5. GC-MS Analysis of Liquid Products

GC-MS analysis revealed that after the hydrochloric acid pretreatment, several small peaks before 30 min disappeared, indicating that the pretreatment could reduce small molecule compounds (<C9) in liquid products (Figure 5a). Hydrochloric acid pretreatment reduced the height of peaks of hexadecanenitrile and hexadecanamide (Figure 5a). Adding metal-oxides increased the height of the peaks belonging to hexadecanenitrile and hexadecanamide irrespective of metal-oxide applied, however, the peak of hexadecanoic acid declined when calcium oxide and magnesium oxide were used (Figure 5b).

The composition of liquid products is shown in Figure 6. The details of the components in bio-oil are listed in Appendix A. These components were classified into five categories: carboxylic acids, hydrocarbon compounds, aromatic compounds, nitrogen compounds, and oxygen compounds.

As shown in Figure 6, after the hydrochloric acid pretreatment, the content of carboxylic acid compounds increased. For the NA sample, carboxylic acid compounds accounted for 41.14%, and increased to 48.52% (1 M) and 51.62% (2 M), respectively. For the 4 M, 6 M, and 8 M samples, it further increased to approximately 63%. Among the carboxylic acid compounds, hexadecanoic acid was the major component. From the data in Appendix A we could learn that after hydrochloric acid pretreatment, the content of the hexadecanoic acid was notably promoted, meaning that its selectivity would also obviously be improved, from 22.4% (NA) to 58.8% (4 M).

On the other hand, ash also had a great impact on the hydrocarbon compounds. With the increase in the concentrations of hydrochloric acid, the relative contents of the three kinds of hydrocarbon compounds (namely heptadecane, 9-hexyl-heptadecane and 5,8-diethyl-dodecane) in the liquid product all increased at first and then decreased. The total content jumped from 12.90% (NA) to 26.67% (2 M), and then fell to 15.42% (8 M).

### 2.6. GPC Analysis of the Bio-Oil

The results of the molecular weight distribution of products including oligomers are shown in Appendix A. According to the data, four parts are divided: 256–400 Da, 400–600 Da, 600–800 Da, and more than 800 Da. For NA samples, the weight distribution of species in 256–400 Da was 66.3%. When pretreated with water (0 M) and dilute acid (0.1 M), the weight distribution of species in 256–400 Da hardly changed. However, for the 1 M, 2 M, 4 M, 6 M, and 8 M samples, the weight distribution of species in 256–400 Da all presented an obvious change, increasing at first and then decreasing. Among them, the highest weight distribution of species in 256–400 Da of 75.3% was achieved for the 4M sample. This indicated that a proper concentration of hydrochloric acid pretreatment was beneficial to promote the degradation of macromolecules in microalgae and inhibit the repolymerization of small molecules to form oligomers. After hydrochloric acid pretreatment, the molecular weight distribution of the other three parts (400–600 Da, 600–800 Da, and greater than 800 Da) were all reduced.

### 2.7. The Characteristics of Gaseous Products

The gas products obtained via pyrolysis were analyzed by GC-TCD as listed in Figure 7. Gaseous products mainly included H_2_, CO, CO_2_, and CH_4_, among which CO_2_ was the main constituent. As presented in Figure 7, the content of CO_2_ declined to some extent after acid pretreatment. Instead, the content of CO was improved. As for the CH_4_ content, it was improved to some extent with the acid pretreatment, indicating that the removal of metals in ash facilitated the pyrolysis of the microalgae in some degree.

## 3. Discussion

Acid pretreatments were preferred for their low energy demands and the ease of scalability. The present work showed the potential of improving transformation efficiency of the natural microalgae from Taihu Lake. The ANOVA analysis, followed by appropriate post hoc test, indicated that the influence of acid concentration on ash content, carbohydrate remaining in solids, and the liquid yield of pyrolysis was significant (all *p* < 0.001) (Appendix A). In addition, the highest removal of ash and different metals (Ca, Mg and Al), the relatively high preservation of carbohydrate in solids and the relatively high yield of liquid in subsequent pyrolysis were obtained from the treatment with 2 mol/L hydrochloric acid on NA, which meant that the 2 mol/L hydrochloric acid had the capability to remove most of the ash. However, Liu et al. [26] found that 0.1 mol/L hydrochloric acid performed the best in pretreating the feedstock and Liu et al. [25] found that 4 mol/L hydrochloric acid was the best concentration to pretreat the feedstock. Thus, for different feedstocks, the optimal concentration for the pretreatment might be different.

The alkaline-earth metals or the transition metals played a crucial role in the behavior of biomass pyrolysis [31,32,33,34]. Therefore, the removal of ash affected the pyrolysis behaviors of the microalgae. The results of TG analysis (Figure 3a) showed that the residual carbon and minerals varied as the concentration of hydrochloric acid increased, so did the yields of pyrolysis products (Figure 4). For the 2 M sample, it achieved the most liquid products because of the removal of metal. The reason might be that the appropriate concentration of hydrochloric acid was able to remove most of the metal-oxides, thereby avoiding the complex catalytic behaviors of these oxides, which in turn enhanced the pyrolysis efficiency. This was consistent with the previous research by Wang [35], where the increase in bio-oil yield due to the removal of some inorganic substances was reported.

It had been proved that an acid solution was able to dissolve the carbohydrates [36,37]. From Figure 2, when the concentration of hydrochloric acid was lower than 2 mol/L, the content of carbohydrates in solid phase hardly changed. However, when the concentration of hydrochloric acid was more than 2 mol/L, the content of carbohydrates in solid phase showed a rapid drop. The variation trend of the content of carbohydrates in liquid phase was contrary to that in solid phase. Therefore, for the sake of maintaining the organic substances in the material, it is better to avoid the dissolution conversion of carbohydrates. The dissolution of carbohydrates also had effect on the efficiency of removing ash during the pretreatment process (Figure 1). The reason might be that more concentrated hydrochloric acid would dissolve more carbohydrates instead of removing metals in ash. The dissolved carbohydrates might react with metals in ash to form some insoluble substances, making it more difficult to remove the metals in ash.

Relevant literature showed that the amides (mainly hexadecanamide) in bio-oils were mainly derived from the secondary reaction, between the carboxylic acid products (hexadecanoic acid) and ammonia gas generated by the pyrolysis of protein. After dehydration of the amide, nitriles (hexadecanenitrile) were further obtained [28,29]. Therefore, it was speculated that the decrease of hexadecanenitrile and hexadecanamide (Figure 5a) might be due to the removal of metal-oxides in ash, which could catalyze the reaction of hexadecanoic acid and ammonia gas to produce more hexadecanenitrile and hexadecanamide. To verify the above deduction, three metal-oxides (calcium oxide, magnesium oxide and aluminum oxide) that presented the most obvious changes in ash content were selected and mixed with 2 M samples for catalytic pyrolysis experiment. Among them, since calcium oxide and magnesium oxide were alkaline oxides, the peak of hexadecanoic acid weakened more obviously. The increasing height of peaks of hexadecanenitrile and hexadecanamide (Figure 5b) provided evidence for the catalytic effects of the metal-oxides in ash. Similar results were reported in the literature [38,39,40].

The composition of liquid products (Figure 6) showed that the relative content of hexadecanoic acid increased after removing metal-oxides in ash. More hexadecanoic acid was obtained from the pyrolysis of the 4 M sample than that from the 2 M sample, this might be caused by the fact that in the solid material, the 4 M sample had less carbohydrates than the 2 M sample. Hexadecanoic acid was an important value-added molecule, which could be used as an additive for food, cosmetic, and pharmaceutical purposes [41]. If hexadecanoic acid could be effectively produced, it would provide a feasible way for the utilization of microalgae.

The results of GPC analysis showed that after acid pretreatment, the weight distribution of species in 256–400 Da increased, and the other three parts (400–600 Da, 600–800 Da, and greater than 800 Da) were all reduced. The reason might be that after the acid pretreatment, on the one hand, the amount of carbohydrates decreased, thus decreased amounts of the products from their conversion could be obtained, including a decrease in oligomers. On the other hand, the carbohydrates that remained in the raw material might be more easily converted to small molecules by pyrolysis.

As Beneroso, D. et al. [42] reported, H_2_ mainly came from the cracking of condensable products and the gasification of carbon. CH_4_ originated from cleavage and depolymerization reactions. CO_2_ was derived from decarboxylation reaction, and CO was obtained from decarbonylation reaction and CO_2_ gasification reaction. Therefore, the declining content of CO_2_ (Figure 7) might be due to the removal of metals from ash, which inhibited the decarboxylation reaction of hexadecanoic acid. The decrease in carbohydrates might also result in the reduction of CO_2_ content. While the increasing content of CO (Figure 7) might be attributed to the decarbonylation of some oxygen compounds.

Table 3 showed the HHV of bio-oil from various green microalgae. Though the HHV of bio-oil from Taihu microalgae was a bit higher than some others, it was needed to consider developing catalytic pyrolysis based on the acid pretreatment to further enhance the pyrolysis behavior and improve the HHV of bio-oil in our future work.

Hydrochloric acid, as a strong inorganic acid, has some disadvantages, such as its corrosiveness to equipment and harm to the environment during the industrial pretreatment process. Meanwhile, because of the presence of the chlorine element, chloroalkanes could be formed in the obtained bio-oil during pyrolysis, which could be detrimental to the utilization of this pretreatment method. However, according to Section 2.2, carbohydrates were detected in the liquid phase, which may have the potential to be further utilized if treated properly. This should also be important as a focus of study in future research.

## 4. Conclusions

The results showed that the liquid yields of pyrolysis increased from 34.4% (NA) to 40.5% (2 M), indicating the inhibition caused by the ash in the pyrolysis process. After acid pretreatment, because of the removal of metal-oxides, the liquid products contained a greater quantity of hexadecanoic acid, while the amount of hexadecanamide and hexadecanenitrile was significantly reduced. The metal-oxides, such as CaO, MgO, and Al_2_O_3_, were capable of catalyzing the reaction of hexadecanoic acid and NH_3_ to produce more hexadecanamide and hexadecanenitrile. In addition, the relative content of hydrocarbon compounds in liquid product was improved from 12.90% (NA) to 26.67% (2 M). The selectivity to hexadecanoic acid was also enhanced by acid pretreatment. For natural microalgae, acid pretreatment at room temperature before pyrolysis could upgrade the bio-oil and efficiently utilize the biomass.

## 5. Materials and Methods

### 5.1. Materials

The natural microalgae (NA), mainly consisted of cyanobacteria *Microcystis*, were collected from Taihu Lake (Wuxi, China) in autumn 2020. The feedstocks were washed with water to removing surface dust. For drying, 10 kg of microalgae were put on a bamboo mat and dried in sunshine for 3 days. Before use, the microalgae were triturated into powder (<0.18 mm) by a sealing pulverizer and dried overnight in an oven at 100 °C. All chemicals used in our work, including ethanol used as solvent for GC-MS analysis and washing fluid for oil attached to both the condenser and the outer tube, hydrochloric acid used for pretreatment, and the metal-oxides (calcium oxide, magnesium oxide, and aluminum oxide) used for catalytic pyrolysis, were purchased from Chron Chemical Company (Chengdu, China). They were all analytical grade.

### 5.2. Pretreatment

20 g microalgae and 200 mL deionized water or HCl solution were put in a 500 mL beaker, which was then set in a shaker (BSD-YX-2000, Boxun, China) at 25 °C for 12 h. The concentrations of HCl were 0.1 mol/L, 1 mol/L, 2 mol/L, 4 mol/L, 6 mol/L, and 8 mol/L, and the obtained samples were noted as 0.1 M, 1 M, 2 M, 4 M, 6 M, and 8 M, respectively. The sample washed with deionized water was named as 0M. Afterwards, all the treated feedstocks were frozen at −80 °C for 4 h and then a freeze-drier (BTP-8ZLE0X, SP SCIENTIFIC, Warminster, PA, USA) was used to freeze-dry the feedstocks at −105 °C and 80 mT for 24 h to remove moisture. The obtained dried samples were all stored in a desiccator.

### 5.3. Pyrolysis Experiment

The pyrolysis experiments were conducted in a fixed-bed reactor, the schematic diagram of which can be found in our previous work [48]. In each run, 0.3 g feedstock was put in the inner tube, which was inserted into the appropriate position of the outer tube, then the thermocouple was tied with the outer tube and the system was placed in the tube furnace. The outer tube was connected to the condensing system with a rubber hose, and the condenser (two coils) was placed in an ice water bath. It was found that after the reaction, little condensation was present in the second coil, indicating that most of the condensable species were collected by the condensation system. The end of the condenser was connected with an air bag to collect gas products.

In this study, the carrier gas was N_2_ and the flow rate was 80 mL/min. According to the results of thermogravimetric analysis, the major weight loss was about 50 wt% from 200 °C to 500 °C. Therefore, we chose 500 °C as the final pyrolysis temperature. The feedstock was heated from 30 °C to 500 °C at a heating rate of 10 °C/min and pyrolyzed for 2 h, to ensure that all the species that can be pyrolyzed were totally converted [6]. The weight difference of the inner tube before and after the reaction was the weight of the solid residue, and the liquid product was the sum of the weight difference of the outer tube and the condenser before and after the reaction. The gas product yield was calculated by the difference method. After the reaction, ethanol was used to wash both the condenser and the outer tube to collect liquid products and dilute the sample to 50 mL for further testing. Every experiment was repeated three times.

In each catalytic pyrolysis experiment, 0.3 g de-ashed microalgae (2 M sample) and 0.15 g metal-oxide were ground together in an agate mortar. The catalytic pyrolysis experiment was carried out in the same reactor under the same conditions as the direct pyrolysis.

The pyrolytic products yields were calculated by Equations (1)−(3),
(1)$$Y_{\mathrm{liquid}}=\frac{{{(W}_{\mathrm{ot}}{+W}_{\mathrm{cond}})}_{\mathrm{after}}-{{(W}_{\mathrm{ot}}{+W}_{\mathrm{cond}})}_{\mathrm{before}}}{W_{\mathrm{sample}}}\times 100\%$$
(2)$$Y_{\mathrm{char}}=\frac{W_{\mathrm{char}}}{W_{\mathrm{sample}}}\times 100\%$$
(3)$$Y_{\mathrm{gas}}{=100\%-Y}_{\mathrm{char}}-Y_{\mathrm{liquid}}$$
where the sum of the weights of outer tube and connecting tube was represented by W_ot_, and the weight of condenser was represented by W_cond_, “before” and “after” stood for the weights of outer tube, connecting tube, and condenser before and after the reaction.

### 5.4. Analysis Method

#### 5.4.1. Characterization of Raw and Pretreated Feedstocks

To determine the metal content in ash of the raw and pretreated feedstocks, inductively coupled plasma-emission spectroscopy (ICP-AES) was used. A 0.2 g sample was weighed and put in beaker, and then 20 mL of HNO_3_ were added. The system was heated to dissolve the sample, and the solution was volatilized until about 3 mL left. Then the solution was diluted to 100 mL with deionized water for ICP analysis using an IRIS Advantage (TJA Solutions, Waltham, MA, USA).

Ultimate analysis: A FLASH 1112SERIES Element Analyzer was employed to determine the contents of carbon, hydrogen, and nitrogen in the raw and acid pretreated feedstocks. The samples were all dried at 80 °C under vacuum overnight before the test.

Proximate analysis: A national standard method GB/T 28731–2012 was implemented. Specifically, the moisture of raw feedstock was measured by weight difference of the sample before and after drying in an oven at 105 °C, until the quantity remained constant. Ash content of the raw and pretreated feedstocks were determined by the weight difference of the sample before and after calcination in a muffle furnace at 550 °C in air, until the quantity remained constant. Volatile content was measured by the weight difference of the samples before and after calcination in a muffle furnace at 900 °C for 7 min. Fixed carbon content was calculated by the equation $\mathrm{FC}\;=100-\left({M\;}_{\mathrm{moisture}}+{\;A\;}_{\mathrm{ash}}+{\;V}_{\mathrm{volatile}}\right)\;$ [9].

Components analysis: Lipids content was analyzed by the method of Bligh and Dyer [49]. The Kjeldahl method was used to determine the proteins content [50]. A DNS colorimetric analysis was used to analyze the content of carbohydrates and a UV 4100-spectrophotometer was employed [51].

Thermogravimetric analysis (NETZSCH STA 449 F5, Germany) was used to analyze the raw and pretreated samples. Specifically, an approx. 10 mg sample was added in an alumina crucible. To prevent the instrument from being polluted by the small particles formed by samples in the analysis, the crucible was covered by a platinum crucible. At a heating rate of 10 °C/min, the sample was heated up to 800 °C. The sweep gas (N_2_) was at the flow rate of 60 mL/min and the protective gas (N_2_) was at the flow rate of 20 mL/min, ensuring that a pure inert atmosphere could be obtained and the gaseous products were able to be moved away from the heating zone.

#### 5.4.2. Characterization of the Liquid Products

An Agilent 5973-6890N gas chromatography/mass spectrometer (GC-MS), equipped with HP-INNOWAX capillary column (30 m × 0.23 mm × 0.25 µm) was used to analyze the bio-oil components. The relative contents of the identified components were calculated by the peak areas. The carrier gas (helium) was at a flow rate of 1.0 mL/min. Both the injector temperature and the detector temperature were set as 280 °C. The GC oven was first maintained at 40 °C for 6 min, heated up to 210 °C at a rate of 5 °C/min, then ascended to 240 °C at a rate of 10 °C/min, and kept 240 °C for 10 min.

A gel permeation chromatograph (GPC, ACQUITY, Waters), equipped with two columns (ACQUITY APC XT 125 2.5 µm & ACQUITY APC XT 45 1.7 µm, 150 mm × 4.6 mm, Waters) and an RID (2414, Waters), was used to analyze the molecular weight distribution of the liquid products. The eluent, THF (HPLC grade), was at a flow rate of 0.6 mL/min. The temperatures of both the detector and column were kept at 50 °C. Empower was used to control the instrument and analyze the data. Specifically, the removal of the ethanol solvent from the liquid product obtained an approx. 2 mg sticky bio-oil sample, which was totally dissolved in 1 mL of THF (HPLC grade), and then filtered by nylon 66 ultrafiltration membrane (with aperture of 0.22 µm). The injection volume was set as 30 µL. According to polystyrene standards in different molecular weights, calibration curves were obtained, which was used to calculate the distribution of molecular weight of the bio-oil. The lowest molecular weight of authentic samples was 226 Da, and the lowest analyzed molecular weight distribution was 256 Da. Therefore, only compounds with Mn > 256 Da were considered in the analysis.

The contents of carbon, hydrogen, and nitrogen in the bio-oil were determined as detailed in Section 5.3. The samples were all dried at 40 °C under vacuum overnight before test.

#### 5.4.3. Characterization of the Gaseous Products

GC-TCD was used to determine the relative contents of gaseous products, which were collected by a gas bag during the whole pyrolysis process. The samples were tested by GC9710 (Fuli) equipped with a TCD detector and a TDX-1 carbon molecular sieve packed column (2 m × 3 mm id) [52]. The main chromatographic parameters were as follows: carrier gas, nitrogen; carrier gas flow rate, 20 mL/min; inlet temperature, 145 °C; column temperature, 120 °C; detector temperature, 180 °C.

### 5.5. Statistical Analysis

The one-way analysis of variance (ANOVA) analyses between acid concentration and variations, including ash content, different metal contents (Ca, Mg and Al), carbohydrate remaining in solids, and liquid yield of pyrolysis, were performed by SPSS (IBM, SPSS statistics, Version 21.0), followed by an appropriate post hoc test (Student–Newman–Keuls). A value of *p* < 0.05 was considered statistically significant.

## Figures and Tables

**Figure 1 toxins-13-00542-f001:**
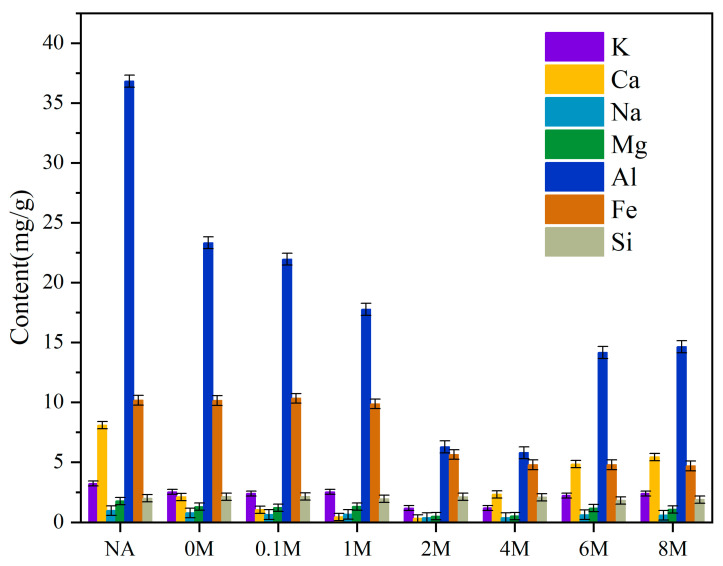
The content change of metals in the materials before and after pretreatment with different concentrations of hydrochloric acid.

**Figure 2 toxins-13-00542-f002:**
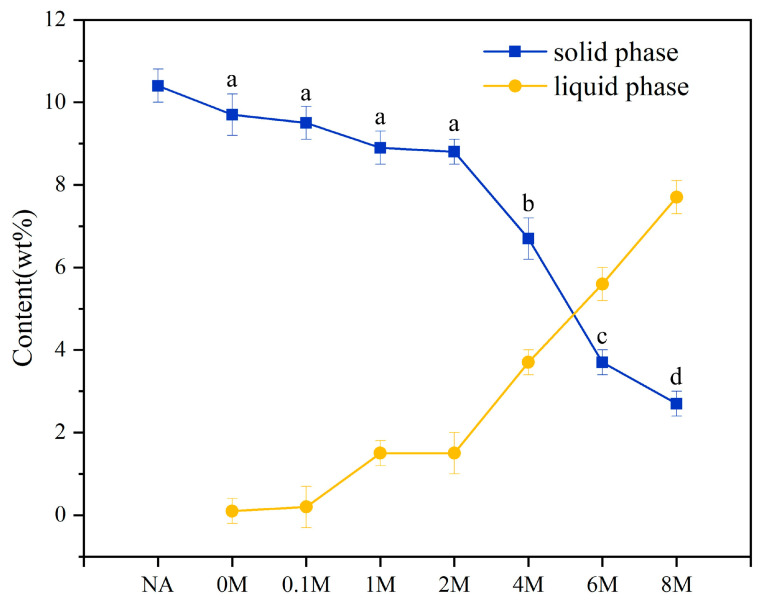
The content of carbohydrates in solid phase and liquid phase before and after hydrochloric acid pretreatment. NA = natural microalgae. The a, b, c, and d were the results of a Student–Newman–Keuls test of ANOVA analysis.

**Figure 3 toxins-13-00542-f003:**
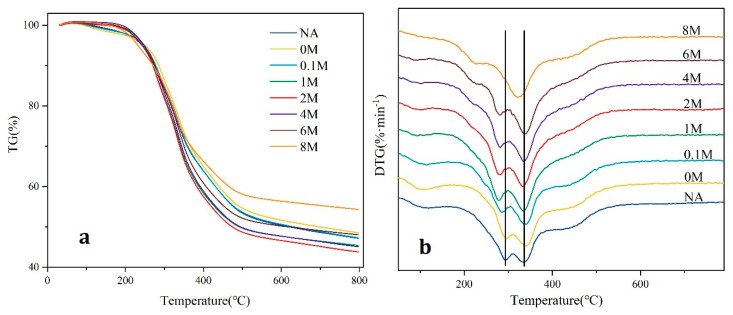
TG analysis of pretreated and untreated samples: (**a**) Thermogravimetric (TG) curves; (**b**) Differential thermogravimetric (DTG) curves.

**Figure 4 toxins-13-00542-f004:**
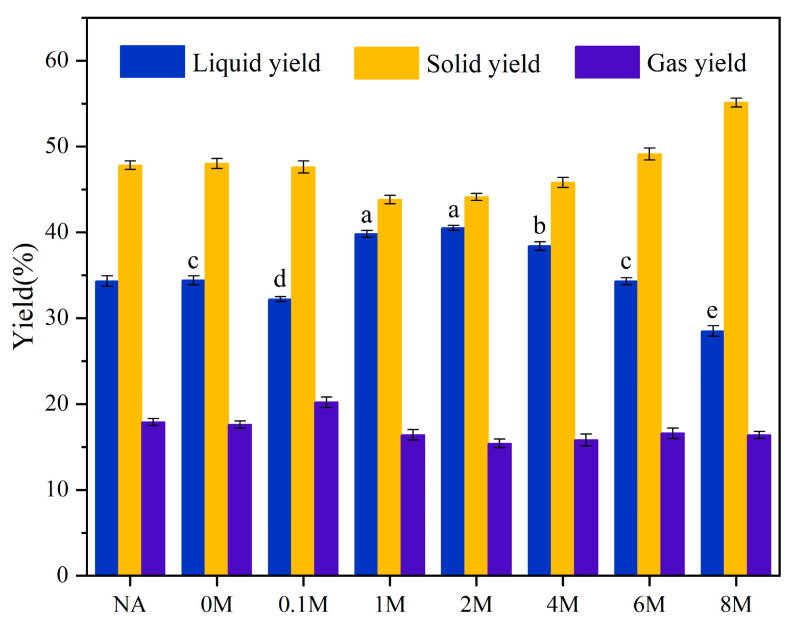
Yields of liquid, solid, and gas products after pyrolysis. NA = natural microalgae. The a, b, c, d, and e were the results of a Student–Newman–Keuls test of the ANOVA analysis.

**Figure 5 toxins-13-00542-f005:**
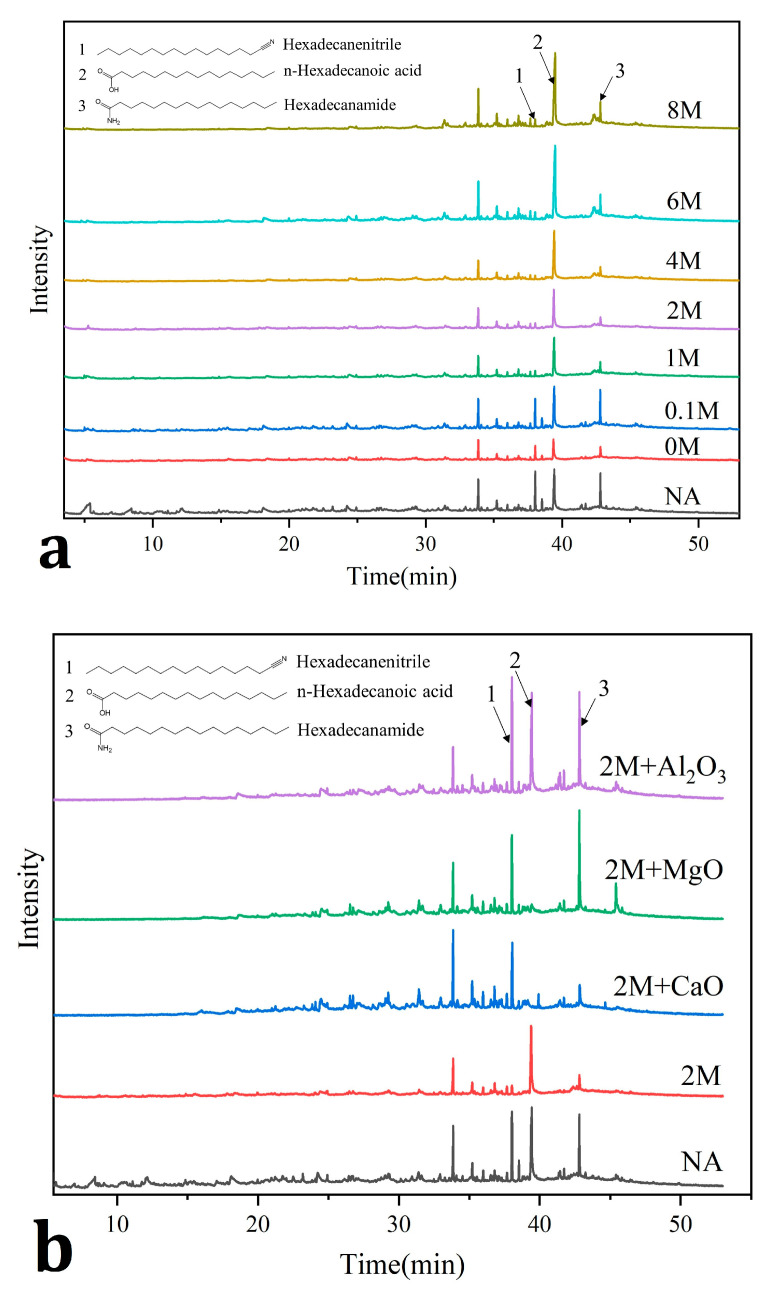
GC-MS analysis: (**a**) GC-MS of liquid products by pyrolysis; (**b**) GC-MS of liquid products by catalytic pyrolysis.

**Figure 6 toxins-13-00542-f006:**
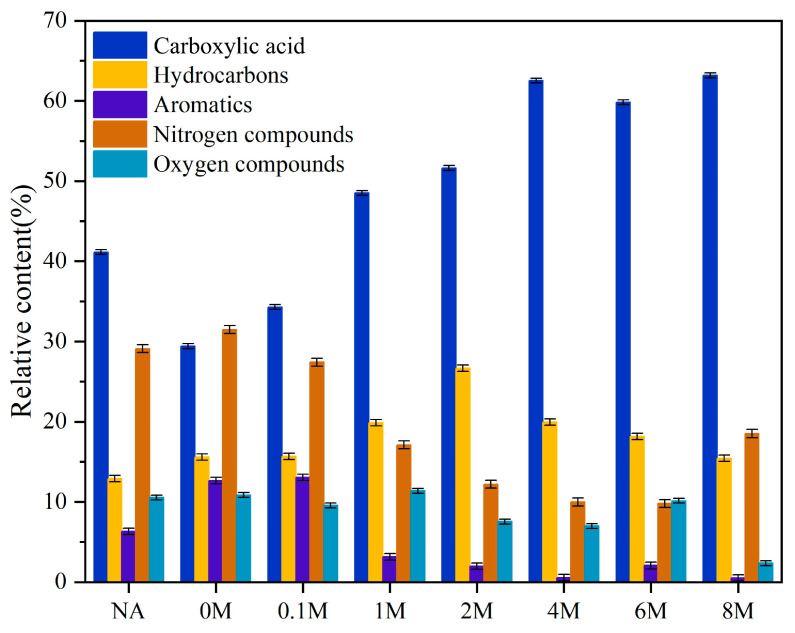
The composition of liquid products by pyrolysis.

**Figure 7 toxins-13-00542-f007:**
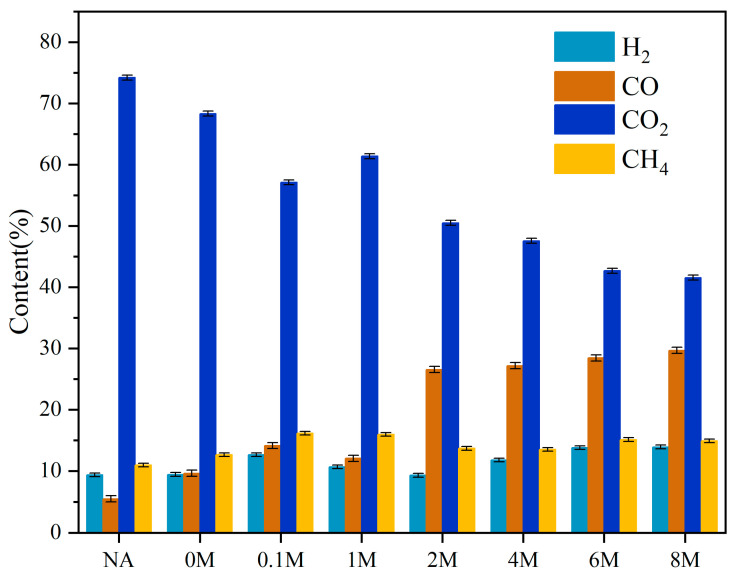
The composition of each gas in gas products (%).

**Table 1 toxins-13-00542-t001:** Ultimate, proximate, components analysis of the raw microalgae.

Ultimate Analysis (wt%)		Proximate Analysis (wt%)		Components Analysis (wt%)	
C	33.01 ± 0.05	Ash	33.2 ± 0.3	Carbohydrates	10.4 ± 0.4
H	5.14 ± 0.02	Moisture	5.8 ± 0.3	Proteins	41.2 ± 0.5
N	6.59 ± 0.03	Volatile matter	45.6 ± 0.2	Lipids	6.6 ± 0.3
O ^a^	22.06 ± 0.02	Fixed carbon	15.4 ± 0.2	Others ^b^	8.6 ± 0.3
HHV (MJ/kg) ^c^	14.90 ± 0.04				

^a^ Calculated by difference, O = 100−C-H-N-Ash; ^b^ Others = 100−Carbohydrates-Proteins-Lipids-Ash; ^c^ HHV = (3.55 C^2^ − 232 C − 2230 H + 51.2 C × H + 131 N + 20,600) × 10^−3^.

**Table 2 toxins-13-00542-t002:** The ash content of the feedstocks before and after pretreatment with different concentrations of hydro-chloric acid.

	NA ^a^	0 M	0.1 M	1 M	2 M	4 M	6 M	8 M
Ash content (wt%)	33.2 ± 0.1	33.7 ± 0.4	33.6 ± 0.5	28.6 ± 0.3	21.1 ± 0.5	30.9 ± 0.5	32.9 ± 0.2	43.3 ± 0.3

^a^ NA = natural microalgae.

**Table 3 toxins-13-00542-t003:** The HHV of bio-oil from various green microalgae.

S.no.	Algae Species	HHV MJ/kg	Temperature, Conditions	Reference
1	Taihu microalgae (Cyanobacteria)	30.4	500 °C, slow pyrolysis	-
2	*Arthrospira platensis*	15.8	550 °C, slow pyrolysis	[43]
3	*Chlamydomonas reinhardtii*	13	350 °C, fast pyrolysis	[44]
4	*Scenedesmus dimorphus*	28.5	500 °C, slow pyrolysis	[45]
5	*C. vulgaris* remnants	24.6	500 °C, fast pyrolysis	[46]
6	*Nannochloropsis* sp. residue	24.4	400 °C, slow pyrolysis	[47]

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
