# Peer review of "Pyrolysis of High-Ash Natural Microalgae from Water Blooms: Effects of Acid Pretreatment"

_toxins, 2021, doi:10.3390/toxins13080542_

Round 1
Reviewer 1 Report
Paper Title: Pyrolysis of high-ash natural algae from water blooms: effects of avid pretreatment
General
The problem was well defined and explained. It was noted why pyrolysis is a good technology choice and why bio-oil has significance. The importance of pretreatment was highlighted along with the preferred method. The methodologies employed were explained. One could follow the work.
The use of algae for producing chemicals, fuels, etc., is very relevant and a top research priority, especially with CO2 uptake to grow the algae. Generally, mass fraction should not be indicated as wt%, only as a mass percent. This should be changed throughout the document.
Observations
Line |
comment |
41 |
Explain transesterification |
42 |
Explain thermochemical routes |
52 |
The suitable pretreatment…. change to A suitable … |
92 |
FID is not defined anywhere |
93 |
Input in parentheses the chemical formula for hexadecenoic acid |
94 |
Input in parentheses the chemical formula for hexadecanamide |
98 |
Change were to are |
119 |
Change to …..cause their content to increase. |
121-122 |
They also augmented with further increase…. the comment is not true for Fe |
145 |
Delete the word While and start the new sentence as The content …. |
149 |
resulting in the increase…. |
150 |
6M,8M… add space after the comma |
155 |
TG and DTG are not explained anywhere in text: cannot only be noted in the figure caption |
171-174 |
Not all statements could be verified; please rewrite |
207 |
Peaks of hexadeconitrile and hexadecanamide need to be indicated in Figure 5a |
212 |
..guessed… use a more scientific word |
219-220 |
Indicate hexadeconitrile and hexadecanamide peaks in Figure 5b; cannot follow the statements |
275 |
Change to carbohydrates that remained |
345 |
Section 5. Materials and methods must be moved after the introduction before the results. It makes no sense at the end, as terms discussed throughout the paper come before the end. |
353 |
…respectively . Remove the space before the period; also, do not start the following sentence with 0M |
Summary
The topic is of high importance for today’s energy transition studies worldwide. Most of the graphs could be followed, with only a few need to be updated and improved. The English language needs another check. When writing units, put the unit after each number or after a parenthesis enclosing both numbers. Minor improvements are required.
Author Response
Responses to Reviewer #1
General
The problem was well defined and explained. It was noted why pyrolysis is a good technology choice and why bio-oil has significance. The importance of pretreatment was highlighted along with the preferred method. The methodologies employed were explained. One could follow the work.
The use of algae for producing chemicals, fuels, etc., is very relevant and a top research priority, especially with CO2 uptake to grow the algae. Generally, mass fraction should not be indicated as wt%, only as a mass percent. This should be changed throughout the document.
Response: Thank you for the comments. Using “wt%” to indicate mass fraction is indeed inaccurate, so we have change “wt%” to “%” in section 2.4, and revised the relevant parts throughout the whole manuscript.
Observations
- Line 41: Explain transesterification
Response: Thank you for the comments. We have added the explanation of transesterification in the manuscript according to literature [1].
- Line 42: Explain thermochemical routes
Response: Thank you for the comments. We have added the explanation of thermochemical routes in the manuscript according to literature [2, 3].
- Line 52: The suitable pretreatment…. change to A suitable …
Response: Thank you for the comments. We have changed “The suitable pretreatment” to “A suitable pretreatment”, and revised the relevant part in the manuscript.
- Line 92: FID is not defined anywhere
Response: Thank you for the comments. We are very sorry that GC-FID was not used, and it is a clerical error. Now it has been deleted and the relevant part in the manuscript has also been revised.
- Line 93: Input in parentheses the chemical formula for hexadecenoic acid
Response: Thank you for the comments. We have added the chemical formula for hexadecanoic acid (C15H31COOH) in the sentence.
- Line 94: Input in parentheses the chemical formula for hexadecanamide
Response: Thank you for the comments. We have added the chemical formula for hexadecanamide (C15H31CONH2) in the sentence.
- Line 98: Change were to are
Response: Thank you for the comments. We have corrected this mistake in grammar; we have also checked throughout the whole manuscript and revised the relevant mistakes.
- Line 119: Change to …..cause their content to increase.
Response: Thank you for the comments. We have changed “cause their content increased” to “cause their content to increase”, and revised the relevant part in the manuscript.
- Line 121-122: They also augmented with further increase…. the comment is not true for Fe
Response: Thank you for the comments. We have revised the former sentence as “The content of Al also augmented with further increase in the concentration of hydrochloric acid”.
- Line 145: Delete the word While and start the new sentence as The content ….
Response: Thank you for the comments. We have revised the former sentence as “The content of carbohydrates in liquid phase increased remarkably for the 4M, 6M and 8M samples”.
- Line 149: resulting in the increase….
Response: Thank you for the comments. We have revised the former sentence as “resulting in the increase in the ash content and metal content for the 4M, 6M, 8M samples”.
- Line 150: 6M,8M… add space after the comma
Response: Thank you for the comments. We have added a space between these two words.
- Line 155: TG and DTG are not explained anywhere in text: cannot only be noted in the figure caption
Response: Thank you for the comments. TG = thermogravimetric and DTG = differential thermogravimetric. We have added the explanation of these two abbreviations in section 2.3.
- Line 171-174: Not all statements could be verified; please rewrite
Response: Thank you for the comments. To make this part more accurate and understandable, We have rewrite these statement in the corresponding part: For the production of residual carbon, NA, 0M and 0.1M samples showed nearly no difference, indicating that water or very dilute acid pretreatment showed little effect on the yield of residual carbon (about 48 %). With the increase of the concentration in pretreatment, the residual carbon started to decrease and reached the least yield for 2M sample (43.9 %). However, as the HCl concentration further increased, the residual carbon started to rise and reached the highest yield for 8M samples (54.4 %). These results showed the same trend as the variation of ash content in Table 2, indicating the positive correlation between the yield of residual carbon and ash content; while the decrease of the peak temperature at 250-290 °C by dilute acid pre-treatment could be one of the reasons for the improved pyrolysis efficiency for 2M sample. These results also indicated that pretreatment by 2 mol/L hydrochloric acid was the best for utilizing pyrolysis process to obtain bio-oil with higher yield.
- Line 207: Peaks of hexadeconitrile and hexadecanamide need to be indicated in Figure 5a
Response: Thank you for the comments. We have added the marks of hexadeconitrile, hexadecanoic acid, and hexadecanamide in Figure 5a.
- Line 212: ..guessed… use a more scientific word
Response: Thank you for the comments. The word “guessed” has now been revised into “speculated”.
- Line 219-220: Indicate hexadeconitrile and hexadecanamide peaks in Figure 5b; cannot follow the statements
Response: Thank you for the comments. We have added the marks of hexadeconitrile, hexadecanoic acid, and hexadecanamide in Figure 5b, and now the statements became more understandable.
- Line 275: Change to carbohydrates that remained
Response: Thank you for the comments. We have revised the sentence as “On the other hand, the carbohydrates that remained in the raw material might be easier to be converted to small molecules by pyrolysis”.
- Line 345: Section 5. Materials and methods must be moved after the introduction before the results. It makes no sense at the end, as terms discussed throughout the paper come before the end.
Response: Thank you for the comments. According to the templates given by the publisher, the section of materials and methods is needed to be set at the end of the article. To make this work more understandable, we have added explanations of abbreviations when they appear for the first time in the article.
- Line 353: …respectively . Remove the space before the period; also, do not start the following sentence with 0M
Response: Thank you for the comments. We have deleted the needless space in this sentence. We also revised the sentence “0M referred to the one washed by deionized water” into “The sample washed by deionized water was named as 0M”.
Summary
The topic is of high importance for today’s energy transition studies worldwide. Most of the graphs could be followed, with only a few need to be updated and improved. The English language needs another check. When writing units, put the unit after each number or after a parenthesis enclosing both numbers. Minor improvements are required.
Response: Thank you very much for your comments. We have revised the whole manuscript according to your comments. The use of language has also been checked. We wish the manuscript is now acceptable.
Submission Date
19 June 2021
Date of this review
13 Jul 2021 22:38:38
Reviewer 2 Report
The comprehensive investigation in this study shows a significant discrepancy between the composition of liquid products from pyrolysis of deashed and raw algae. The optimal influence of ash washing with HCl on bio-oil yield and quality is proven, which is beneficial for the reutilization of harmful algae. This research also gives an interesting point to figure out the effect of different metal oxides in the ash. However, there are still some minor points that should be clarified.
Line 171-176: According to Table 1, ash content is the lowest with 2M HCl washing, so that’s why you have the least content of residual carbon and minerals after TGA test in Figure 3a. The temperature for the first peak in DTG can be reduced after washing, so the pyrolysis efficiency is enhanced by this pretreatment which might be a better reason why washing is conductive to the pyrolysis process.
Line 205: Small molecule compounds are not so clear. Maybe you can use the range of carbon number (Cn) to explain.
Figure 8: Is all producer gas during the pyrolysis duration all collected in the gas bag? Or only in a time range? It should be clarified in the section of Material and Methods.
Line 327-331: You can enrich the information on the quality of bio-oil produced in this study by comparing it with different liquid fuels (heating value, acid value, viscosity, etc.). Heating value is only one of the items to evaluate the quality of bio-oil for the application as fuel. Then, readers can know more about its potential for energy purpose.
- Material and Methods: For the experiment design, algae was heated from 30 °C to 500 °C and then pyrolyzed for 2 h. It’s actually a long duration for pyrolysis. The major products might be formed only at the beginning of time. Can you give us some description for why you choose 500 °C and 2h as the main condition for this study, although I know pyrolysis at 500°C is normally for liquid production? In addition, it’s better to have more detail for the description of how you collect liquid samples since bio-oil is the major focus in this study. The design of your sampling system is related to the liquid capture efficiency from the exhaust gas. The system should be reliable.
Author Response
Responses to Reviewer #2
Comments and Suggestions for Authors
The comprehensive investigation in this study shows a significant discrepancy between the composition of liquid products from pyrolysis of deashed and raw algae. The optimal influence of ash washing with HCl on bio-oil yield and quality is proven, which is beneficial for the reutilization of harmful algae. This research also gives an interesting point to figure out the effect of different metal oxides in the ash. However, there are still some minor points that should be clarified.
Response: Thank you for your comments! We have revised he manuscript accordingly, and the point-to-point responses are listed below:
- Line 171-176: According to Table 1, ash content is the lowest with 2M HCl washing, so that’s why you have the least content of residual carbon and minerals after TGA test in Figure 3a. The temperature for the first peak in DTG can be reduced after washing, so the pyrolysis efficiency is enhanced by this pretreatment which might be a better reason why washing is conductive to the pyrolysis process.
Response: Thank you for the comments. We agree with you and we have rewritten these statement in the corresponding part: For the production of residual carbon, NA, 0M and 0.1M samples showed nearly no difference, indicating that water or very dilute acid pretreatment showed little effect on the yield of residual carbon (about 48 %). With the increase of the concentration in pretreatment, the residual carbon started to decrease and reached the least yield for 2M sample (43.9 %). However, as the HCl concentration further increased, the residual carbon started to rise and reached the highest yield for 8M samples (54.4 %). These results showed the same trend as the variation of ash content in Table 2, indicating the positive correlation between the yield of residual carbon and ash content; while the decrease of the peak temperature at 250-290 °C by dilute acid pre-treatment could be one of the reasons for the improved pyrolysis efficiency for 2M sample. These results also indicated that pretreatment by 2 mol/L hydrochloric acid was the best for utilizing pyrolysis process to obtain bio-oil with higher yield.
- Line 205: Small molecule compounds are not so clear. Maybe you can use the range of carbon number (Cn) to explain.
Response: Thank you for the comments. We have added “<C9” to explain the small molecules.
- Figure 8: Is all producer gas during the pyrolysis duration all collected in the gas bag? Or only in a time range? It should be clarified in the section of Material and Methods.
Response: Thank you for the comments. The produced gas was collected in a gas bag in the whole pyrolysis process. We have clarified it in the section of Material and Methods.
- Line 327-331: You can enrich the information on the quality of bio-oil produced in this study by comparing it with different liquid fuels (heating value, acid value, viscosity, etc.). Heating value is only one of the items to evaluate the quality of bio-oil for the application as fuel. Then, readers can know more about its potential for energy purpose.
Response: Thank you for the comments. We agree with you that the comparation of our bio-oil and others’ is necessary, so we have listed the higher heating values of our bio-oil and those in other researches in Table 3, and compared them in the manuscript. However, unfortunately, the bio-oil in our work is collected by washing with ethanol, as was mentioned in the section of Material and Methods, it is difficult to measure the acid value and viscosity of the bio-oil. We would attempt to resolve this in our future work.
- Material and Methods: For the experiment design, algae was heated from 30 °C to 500 °C and then pyrolyzed for 2 h. It’s actually a long duration for pyrolysis. The major products might be formed only at the beginning of time. Can you give us some description for why you choose 500 °C and 2h as the main condition for this study, although I know pyrolysis at 500 °C is normally for liquid production? In addition, it’s better to have more detail for the description of how you collect liquid samples since bio-oil is the major focus in this study. The design of your sampling system is related to the liquid capture efficiency from the exhaust gas. The system should be reliable.
Response: Thank you for the comments. According to the results of thermogravimetric analysis, the major weight loss was about 50 wt% from 200 to 500 °C. Therefore, we chose 500 °C as the final pyrolysis temperature. We have added this part of statement in the section of Material and Methods. As for the duration time of pyrolysis, we chose 2 h according to the previous work to ensure that all the species that can be pyrolyzed were totally converted [4]. We have added this part of statement in the section of discussion. To collect liquid products, we set two coils in ice water bath. It was found that after the reaction, little condensation was found in the second coil, indicating that most of the condensable species were collected by the condensation system. We have revised the relevant part of the manuscript.
Submission Date
19 June 2021
Date of this review
07 Jul 2021 15:10:33
Reviewer 3 Report
Dear Authors,
The MS was well-written and presents an interesting and insightful study.
Good luck
Author Response
Responses to Reviewer #3
Comments and Suggestions for Authors
Dear Authors,
The MS was well-written and presents an interesting and insightful study.
Good luck
Response: Thank you very much for your positive comments.
Submission Date
19 June 2021
Date of this review
15 Jul 2021 09:44:20
Reviewer 4 Report
- You need to revise the article title as “microalgae” due to its different from macroalgae (algae). Did you characterize the microalgae? Which genus was dominant?
- In Line 67, Chlorella should be written in italic.
- In line 102 and Table 1, what is the “NA”, describe and denote the full name.
- You can present Figure 7 in the Supplementation. There seem to not big differences in the different groups.
- There has been a lack of statistical analysis. After that, you should compare each parameter among them.
- How many replications did you analyze? I could not see the standard deviations in Table 1, and Figures 1, 2, 4, 6 and 8.
- Table S4 contains important data for microalgae. You can move it to main manuscript. This table was also lack of standard deviations.
- What is the 373 K (in lines 350, 386 and 430)? You can convert to Celsius because you describe others in °C.
- In line 380, breaker or beaker?
- You can discuss the negative effect of hydrochloric acid on the environment and its corrosive effects.
Author Response
Responses to Reviewer #4
Comments and Suggestions for Authors
- You need to revise the article title as “microalgae” due to its different from macroalgae (algae). Did you characterize the microalgae? Which genus was dominant?
Response: Thank you for the comments. We agree with you that it is necessary to explain the species of the algae, so we have changed “algae” throughout the manuscript to “microalgae”. We have characterized the microalgae, and the results are shown in Table 1. The microalgae is mainly consisted of cyanobacteria Microcystis. We have added this statement in the section 2.1.
- In Line 67, Chlorella should be written in italic.
Response: Thank you for the comments. We have written Chlorella in italic.
- In line 102 and Table 1, what is the “NA”, describe and denote the full name.
Response: Thank you for the comments. NA = natural microalgae. We have added an explanation and a footnote in line 102 and Table 1, respectively.
- You can present Figure 7 in the Supplementation. There seem to not big differences in the different groups.
Response: Thank you for the comments. We agree with you that less differences can be found in Figure 7, so we have now moved it into the Supplementation as Figure S1, and therefore we have adjusted the serial number of the figures as well.
- There has been a lack of statistical analysis. After that, you should compare each parameter among them.
Response: Thank you for the comments. In this work, the conditions, such as temperature, time, carrier gas flow rate and heating rate, were chosen according to previous work. Therefore, this work focused more on the effect of acid pretreatment on pyrolysis process, less on the comparison of the parameter.
- How many replications did you analyze? I could not see the standard deviations in Table 1, and Figures 1, 2, 4, 6 and 8.
Response: Thank you for the comments. Every experiment and characterization were repeated for three times, and we have added this in the section of Materials and Methods. Furthermore, we have added the standard deviations in Table 1 (the former Table S4) and Table 2 (the former Table 1), and the error bars in Figure 1, 2, 4, 6, and 7 (the former Figure 8).
- Table S4 contains important data for microalgae. You can move it to main manuscript. This table was also lack of standard deviations.
Response: Thank you for the comments. We agree with you that Table S4 is significant, so we have now moved into manuscript as Table 1. Standard deviations have been added in Table 1.
- What is the 373 K (in lines 350, 386 and 430)? You can convert to Celsius because you describe others in °C.
Response: Thank you for the comments. We have changed all the unit K into oC in the manuscript.
- In line 380, breaker or beaker?
Response: Thank you for the comments. We wanted to express “beaker”; it is a clerical error and we have revised relevant part in the manuscript.
- You can discuss the negative effect of hydrochloric acid on the environment and its corrosive effects.
Response: Thank you for the comments. We agree with you that the negative effects of hydrochloric acid should be mentioned in the manuscript. Hydrochloric acid, as a strong inorganic acid, has some disadvantages, such as the corrosiveness to the equipment and the harm to the environment during industrial pretreatment process. Meanwhile, because of the presence of chlorine element, chloroalkanes could be formed in the obtained bio-oil during pyrolysis, which could be detrimental to the utilization of this pretreatment method. However, according to section 2.2, carbohydrates were detected to be converted into the filtrates, which may have the potential to be further utilized if being treated properly. This should be important to be studied in the future work. We have added these statements in the section of discussion.
Submission Date
19 June 2021
Date of this review
08 Jul 2021 10:58:35
Round 2
Reviewer 4 Report
The manuscript was well revised, now can be publishable.
Author Response
Dear authors, I have received the request to evaluate your manuscript. As you can read from the reviewers judgements, the reviewers in general are in favour of publication of your work.
I have also read your submission with great interest, especially given the potential of using and adding benefits to otherwise nuisance cyanobacterial biomass. Nonetheless, there are still some issues that need to be addressed before I can recommend publication of your submission.
Your works can be improved by some rewriting and restructuring as well as adding scientific accuracy, such as presenting results only in the results, giving interpretations of results in the discussion, adding more information and detail to the methods, including statistical analysis and using the journal referencing format in references.
Please, find my comments below:
Response: Thank you very much for the comment! We have revised the manuscript according to your comments, and the point-to-point responses to your comments are listed below.
- Line 108, start the results with text instead of a table
Response: Thank you for the comment. We have moved the text to the beginning of the results, and now this section starts with “In this work, the natural microalgae (NA) from Taihu Lake…”.
- Line 118, “...and the oxygen content is high...”, there is no oxygen given in table 1
Response: Thank you for the comment. The oxygen content in this work was calculated by the equation of “100 – C – H – N – Ash”. To clarify the calculation, the term “others” in the former Table 1 has now been changed to “O”, the footnote a was changed from “Others = 100 – C – H – N” to “O = 100 – C – H – N – Ash”.
- Line 132/133, “The alkaline-earth metals or the transition metals played a crucial role on the behavior of biomass pyrolysis [27-30]”. This is an interpretation of the results that belongs to the discussion.
Response: Thank you for the comment. We agree with you that this interpretation of the results belongs to the section of discussion, so we have now moved this sentence to the section of discussion.
- Line 176, Y-axis indicates the mass fraction of carbohydrate, label should be “Carbohydrate content (wt%)”.
Response: Thank you for the comment. We have revised the label of Y-axis in Figure 2 as “Carbohydrate content (wt%)”.
- Line 178, explain what NA means in figure 2 caption
Response: Thank you for the comment. We have added “NA = natural microalgae” to explain its meaning in the caption of Figure 2.
- Line 228/230, there is no need to start with sentences that results are given in graphs/tables, referencing to the corresponding graph/table can be done in text. For instance, “The products distributions are compared in Figure 4. For the liquid yield, it was 34.2 % for the NA sample, and there was almost no change when pretreated with water (0M) or dilute acid (0.1M).” can be rewritten as “The liquid yield in NA samples was on average 34.2 %, and there was almost no change when pretreated with water (0M) or dilute acid (0.1M) (Figure 4).”
Response: Thank you for the comment. We agree with you that this paragraph should be more concise. Therefore, these sentences have been rewritten as “The liquid yield of NA samples was on average 34.2 %, with no obvious changes compared with the results of the samples pretreated with water (0M) or dilute acid (0.1M) (Figure 4).”
- Line 234/236, interpretation belongs to the discussion.
Response: Thank you for the comment. We agree with you that this interpretation belongs to the section of discussion, so the interpretation “The reason might be that the appropriate concentration of hydrochloric acid was able to remove most of the metal oxides, thereby avoiding the complex catalytic behaviors by these oxides, which in turn enhanced the pyrolysis efficiency” has now been moved to the section of discussion.
- Line 240/243, Also this part can be condensed. The original “The liquid products were detected by GC - MS, and the results were shown in Figure 5a. It could be seen that after the hydrochloric acid pretreatment, some small peaks before 30 minutes disappeared, indicating that the pretreatment could reduce some small molecule compounds (< C9) in liquid products” can be replaced with something like “GC – MS analysis revealed that after the hydrochloric acid pretreatment several small peaks before 30 minutes disappeared, indicating that the pretreatment could reduce small molecule compounds (< C9) in liquid products (Figure 5a)”.
Response: Thank you for the comment. We agree with you that this part should be more concise. We have now rewritten these sentences as “GC-MS analysis revealed that after the hydrochloric acid pretreatment, several small peaks before 30 minutes disappeared, indicating that the pretreatment could reduce small molecule compounds (< C9) in liquid products (Figure 5a)”.
- Line 244/252 reads a bit strange in a results chapter. This paragraph can easily be condensed to “Hydrochloric acid pretreatment reduced the height of peaks of hexadeconitrile and hexade-canamide (Figure 5a)”. The sentences referring to literature can be placed in the introduction and/or discussion.
Response: Thank you for the comment. We agree with you that this paragraph should be condensed. Therefore, this paragraph has now been rewritten to “Hydrochloric acid pretreatment reduced the height of peaks of hexadeconitrile and hexadecanamide (Figure 5a)”. Besides, the sentences referring to literature have been moved to discussion part.
- Line 253/256 belongs to the methods where more detail should be provided. Where were metal oxides obtained from, how much was added?
Response: Thank you for the comment. In this work, all the metal oxides were purchased from Chron Chemical Company (Chengdu, China). In each run, 0.3 g microalgae pretreated by 2M hydrochloride solution (2M sample) and 0.15 g metal oxide were grinded together in an agate mortar for further catalytic pyrolysis. We have added these details in materials and methods.
- Line 256/261, again, just report the results, for instance “Adding metal-oxides increased the height of the peaks belonging to hexadeconitrile and hexadecanamide irrespective of metal-oxide applied, however, the peak of hexadecanoic acid declined when calcium oxide and magnesium oxide were used (Figure 5b).”
Response: Thank you for the comment. We agree that this paragraph should be more concise. Therefore, this paragraph has been rewritten as “Adding metal-oxides increased the height of the peaks belonging to hexadeconitrile and hexadecanamide irrespective of metal-oxide applied, however, the peak of hexadecanoic acid declined when calcium oxide and magnesium oxide were used (Figure 5b).”
- Line 257 (and also 382), do not use the word significantly without statistical testing
Response: Thank you for the comment. We agree with you that the word “significantly” is not proper here. The sentence at former line 257 has now been deleted, while in the former sentence at line 382, the statement “significantly reduced” has now been revised as “reduced obviously”.
- Line 262/268, the paragraph doesn’t belong in the results.
Response: Thank you for the comment. We agree with you that this paragraph does not belong to this section, so we have now integrated this paragraph into the section of discussion.
- Line 271, figure 6, legend of first blue should read “carboxylic”.
Response: Thank you for the comment. We have corrected the word “carbonxylic” to “carboxylic”.
- Line 273, this sentence belongs to the methods
Response: Thank you for the comment. We have moved this sentence to section 5.4.2.
- Line 328, the discussion is rather limited and could be extended with parts that are now in the results. In a discussion you reflect on your hypothesis, give interpretations to the results obtained that are placed in context using literature. Discuss all your results.
Response: Thank you for the comment. We have rewritten the section of discussion. It has now been extended with the present results; all the results were discussed.
- Line 330/331, how stable is the composition of the harvested biomass from Lake Taihu? In the methods it is mentioned that biomass was harvested in autumn, but will a 2M treatment also be fine for biomass harvested in spring, or summer?
Response: Thank you for the comment. The natural microalgae in this work was harvested in autumn, while the same feedstock harvested in 2010 summer and 2012 summer were also used in our previous work [1, 2]. Comparing the compositions of the feedstock used in these three works, less variation was found. It is speculated that the composition of the feedstock should be stable enough no matter it is harvested in summer or autumn, so the 2M treatment should also be effective on this natural microalgae harvested in summer. It is indeed meaningful to study whether the same pretreatment method is effective to the microalgae obtained in spring or winter, but in this work, we did not focus our concentration on this aspect. Thanks for your guidance, we may study the effect of the harvest seasons of microalgae on the efficiency of hydrochloride acid pretreatment in our future work.
- Line 361/363, this belongs to the results
Response: Thank you for the comment. We agree with you that this part belongs to the results, so we have moved these sentences to the results of section 2.4.
- Line 392, how was the biomass of Microcystis collected, where, when, how much, how was it stored?
Response: Thank you for the comment. The feedstock used in our work was natural microalgae. It was mainly consisted of cyanobacteria Microcystis and was collected manually from Taihu Lake (Jiangsu, China) in 2020 autumn. It was stored in wild-mouth bottle after drying. We have revised the relevant part of the manuscript.
- Line 393, how was the biomass washed, with what purpose? How was it dried, how much was used, in what kind of containers, for how long?
Response: Thank you for the comment. The harvested biomass was washed by water for removing dust on the surface. For drying, 10 kg microalgae were put on bamboo mat and dried in sunshine for 3 days.
- Line 394, how was the dried matter grinded? Give mesh size in micrometer.
Response: Thank you for the comment. The dried matter was grinded by a sealing pulverizer. The 80 mesh is equivalent to 0.18 mm. We have used micrometer to describe the size of the feedstock in the manuscript and revised the relevant part of the manuscript.
- Line 395, be more specific about “washing”. What was done? How much of dried matter was resuspended in how much liquid? Was this shaken or not?
Response: Thank you for the comment. We have added a new section of “pretreatment” as the section 5.2 to describe the pretreatment process. “20 g microalgae and 200 mL deionized water or HCl solution were put in a 500 mL beaker, then it was set in a shaker (BSD-YX-2000, Boxun, China) at 25 °C for 12 h. The concentrations of HCl were 0.1 mol/L, 1 mol/L, 2 mol/L, 4 mol/L, 6 mol/L, and 8 mol/L, and the obtained samples were noted as 0.1M, 1M, 2M, 4M, 6M, 8M, respectively. The sample washed by deionized water was named as 0M. Afterwards, all the treated feedstocks were frozen at -80 °C for 4 h and then a freeze-drier (BTP-8ZLE0X, SP SCIENTIFIC, USA) was used to freeze-dry the feedtocks at -105 °C and 80 mT for 24 h to remove the moisture. The obtained dried samples were all stored in a desiccator.” Besides, since we added a new minor section, we have also adjusted the serial numbers of the minor sections in section 5.
- Line 400, it reads as if the samples were put in a freeze-drier, clarify what was done.
Response: Thank you for the comment. We indeed used freeze-drier to dry our samples. All the treated feedstocks were frozen at -80 °C for 4 h and then a freeze-drier (BTP-8ZLE0X, SP SCIENTIFIC, USA) was used to freeze-dry the feedtocks at -105 °C and 80 mT for 24 h to remove the moisture. The obtained dried samples were all stored in a desiccator. We have added this statement into the manuscript.
- Line 421, “Every experiment was repeated for three times.” I miss the statistical analysis.
Response: Thank you for the comment. Statistical analysis was performed using the Student’s t-test and differences between NA sample, 8M sample and 2M sample were judged to be significant at *P < 0.05 and very significant at **P < 0.01, ***P < 0.001 and ****P < 0.0001. We have added the statistical analysis to a new Section 5.5 and Section 2.4.
- Line 450/451 can be omitted
Response: Thank you for the comment. We have deleted these sentences.
- References,please use the journal style (see https://www.mdpi.com/journal/toxins/instructions): 1. Author 1, A.B.; Author 2, C.D. Title of the article. Abbreviated Journal Name Year, Volume, page range.
Response: Thank you for the comment. We have revised the journal style of Reference according to the instruction.
Looking forward to your revision, Kind regards
[1] Y. Zeng, J. Tang, S. Lian, D. Tong, C. Hu. Study on the conversion of cyanobacteria of Taihu Lake water blooms to biofuels. Biomass and Bioenergy 2015, 73, 95-101.
[2] R. Zhang, L. Li, D. Tong, C. Hu. Microwave-enhanced pyrolysis of natural algae from water blooms. Bioresour Technol 2016, 212, 311-317.